# Neuropsychological Effects of the Lockdown Due to the COVID-19 Pandemic on Patients with Alzheimer’s Disease and Their Caregivers: The “ACQUA” (Alzheimer–COVID QUArantine Questionnaire) Study

**DOI:** 10.3390/ijerph21121622

**Published:** 2024-12-04

**Authors:** Alessandro Trebbastoni, Roberta Margiotta, Fabrizia D’Antonio, Sonia Barbetti, Marco Canevelli, Sofia Diana, Antonella Di Vita, Letizia Imbriano, Micaela Sepe Monti, Giuseppina Talarico, Cecilia Guariglia, Giuseppe Bruno

**Affiliations:** 1Department of Human Neuroscience, Sapienza University of Rome, 00185 Rome, Italy; roberta.margiotta@uniroma1.it (R.M.); fabrizia.dantonio@uniroma1.it (F.D.); sonia.barbetti@uniroma1.it (S.B.); marco.canevelli@uniroma1.it (M.C.); sofia.diana@uniroma1.it (S.D.); letiziaimbriano1982@gmail.com (L.I.); micaela.sepemonti@uniroma1.it (M.S.M.); giuseppina.talarico@uniroma1.it (G.T.); giuseppe.bruno@uniroma1.it (G.B.); 2Stroke Unit, Department of Emergency, Ospedale dei Castelli, 00040 Ariccia, Italy; 3PhD Program in Behavioral Neuroscience, Sapienza University of Rome, 00185 Rome, Italy; 4ASReM—Azienda Sanitaria Regionale del Molise, 86100 Campobasso, Italy; antonella.divita83@gmail.com; 5Department of Psychology, Sapienza University of Rome, 00185 Rome, Italy; cecilia.guariglia@uniroma1.it; 6Cognitive and Motor Rehabilitation and Neuroimaging Unit, Istituto di Ricovero e Cura a Carattere Scientifico (IRCCS), Fondazione Santa Lucia, 00179 Rome, Italy

**Keywords:** COVID-19, SARS-CoV-2, pandemic, lockdown, quarantine, dementia, Alzheimer’s disease, cognitive impairment, caregivers, questionnaire

## Abstract

Background: The lockdown due to the COVID-19 pandemic, imposed in many countries in 2021, led to social isolation and the interruption of many activities that were useful in stimulating cognition. The impact of these changes has been particularly severe in older subjects with cognitive impairment. Methods: The present study aimed to investigate the effects of lockdown on Alzheimer’s disease patients (in cognition, behavior, and autonomy) and on their caregivers (in emotions, burden, and quality of life). We created a questionnaire and performed an extensive semi-structured telephone interview with each caregiver. The main outcomes were (1) changes in cognitive and behavioral symptoms and autonomy levels in the patients and (2) effects on caregivers’ emotions, burden, and quality of life. Results: The lockdown severely impaired patients’ cognition and independence and worsened behavioral and psychological symptoms of dementia. These effects contributed to increasing caregivers’ burden and stress levels, with a significant perceived deterioration in quality of life among caregivers with higher education levels (*p* = 0.047). Conclusions: This study might contribute to our understanding of the impact of lockdown on Alzheimer’s disease patients and their caregivers, to guide future public health interventions aimed at preventing and/or reducing the consequences of similar extraordinary events in frail subjects.

## 1. Introduction

Severe acute respiratory syndrome coronavirus 2 (SARS-CoV-2) was discovered in Italy in February 2020 and resulted in a series of government restrictions that aimed to manage the spread of the infection. From the 9th of March to the 3rd of May 2020, the Italian government imposed a national lockdown in response to the dramatic increase in COVID-19 cases in the country. Authorities introduced strategies to limit the movement of people throughout the territory, including working from home and online schooling. These unprecedented confinement measures caused a suspension of social and recreational activities, a drastic reduction in visits from family and friends, and a general lack of social contact.

The consequent changes in daily routine could lead to increased anxiety and apprehension in relation to coping with adaptation to a different lifestyle [1]. Furthermore, increased vigilance due to the fear of infecting close relatives and losing family members or friends due to the virus could compromise psychological well-being and mental health [2]. The consequences of such restrictions significantly affected frailer individuals, particularly those with cognitive impairment and fewer coping resources, whose vulnerability could be further exacerbated [3]. Indeed, people with dementia, in particular, Alzheimer’s disease (AD), are among the most vulnerable in society and represent one of the populations at the greatest risk of negative outcomes during quarantine [3]. They depend on the care of family members or non-familial caregivers for the activities of daily living (ADL) and frequently have comprehension, reasoning, and learning impairments that can cause difficulties in adapting to new life circumstances [4]. The inability to understand the reasons for the necessary precautions due to pandemic restrictions or to remember why these changes to daily life have been implemented can cause distress to patients and caregivers.

Moreover, these patients very often show comorbid behavioral and psychological symptoms of dementia (BPSD) [5] that might be exacerbated by forced confinement. Furthermore, reduced mental activation due to the interruption of previous activities, such as attending day-care centers for older people or cognitive stimulation sessions, can significantly contribute to cognitive decline [6] and worsen BPSD. Physical exercise is also a critical element in the well-being of patients with dementia. Movement restrictions may also contribute to general worsening in such patients [7].

The caregivers of a patient suffering from AD very often experience a high burden in their daily care [8]. The prolonged restrictions due to the lockdown imposed in many countries during the COVID-19 pandemic might have represented a further stressful event for these caregivers. We hypothesized that the discomfort that they experienced in a completely new situation, the quarantine, could trigger abnormal reactions, thus creating a pathological loop involving a mutual increase in mental distress between them and their patients.

Our study aimed to investigate caregivers’ perspectives on the effects of the restrictions due to the lockdown on the cognition, behavior, and autonomy levels of patients with a diagnosis of AD through a telephone survey. We also addressed the impact of such extraordinary care experiences on caregivers’ emotions and moods. We hypothesized that the pandemic and its accompanying control measures might worsen patients’ cognitive and behavioral impairment over time, leading to an increase in care needs for those who rely most strongly on personal care. We also hypothesized that such restrictions might negatively influence family caregivers’ physical and mental health, mainly if they force them to increase the frequency and the burden of their care responsibilities.

## 2. Materials and Methods

### 2.1. Participants

One hundred and ninety-seven caregivers aged 29 to 89 (women = 127; median age = 58 years) participated in this study. Participants were consecutively selected among the caregivers of mild-to-moderate AD patients with a Mini Mental State Examination (MMSE) [9] score ranging from 16 to 24 attending the Memory Clinic of the Department of Human Neuroscience, Policlinico Umberto I University Hospital of Rome. This study included only family caregivers responsible for the patients’ day-to-day care before the lockdown. They had to meet some inclusion criteria for participation in the study. First, an eligible caregiver had to be a reliable study partner, in frequent contact with the patient (i.e., at least 15 h per week before the lockdown). Second, they had to be available by telephone at designated times and have adequate literacy to complete the protocol-specified questionnaire (5 years of education at least). Lastly, they had to have accompanied the patient for a visit at least once over the twelve months before the beginning of the COVID-19 pandemic.

### 2.2. Alzheimer–COVID QUArantine (ACQUA) Questionnaire

For the present study, we conceived a specific semi-structured questionnaire: the Alzheimer–COVID QUArantine questionnaire (ACQUA) (see Appendix A). It comprised 157 open and closed questions, with dichotomous or Likert scale answers. Trained neuropsychologists, blinded to the patient’s disease severity, administered the questionnaire by phone to the caregivers enrolled. The interview lasted about 45–60 min. The raters anonymously recorded participants’ answers, excluding any references (streets or names) that could jeopardize participant anonymity. To ensure the complete availability of the caregiver for all the time necessary to complete the interview, we scheduled a preliminary telephone call aimed at confirming the eligibility of the interviewed caregivers by checking the inclusion/exclusion criteria, obtaining their informed consent for the study, and planning a subsequent appointment for the complete survey. We administered the ACQUA questionnaire from August to September 2020, within the first three months of lockdown isolation. The questions included in the survey investigated four different areas of interest (Figure 1).

In the first section (questions 1–27), we collected the demographic data of patients and caregivers: age, gender, education level (years), marital status (categorized as married, living with a partner, single, divorced, or widowed), and primary work status (categorized as working, unemployed, on sick leave, retired, student, or other). A specific question investigated whether the caregiver cohabited or not with the patient before the pandemic. We also asked about participants’ general health conditions related to the occurrence of SARS-CoV-2 infection (symptoms, number of nasopharyngeal swabs carried out and their results, and any medical consultation/pharmacological intervention needed). In the second section (questions 28–48), we extensively studied the daily organization of the caregiving during lockdown compared to before the containment measures taken by the Italian government came into force. There, we focused on the management of patients’ needs related to any major changes in living conditions (moving in or not moving in with the patient, hiring an external caregiver, or increasing the patient’s everyday assistance), and any changes in the frequency of contact with friends and relatives (number of meetings/telephone calls with relatives and/or friends per week). The third part of the questionnaire (questions 49–127) explored the caregivers’ perspective on the effect of lockdown on patients’ cognition, behavior, autonomy, and awareness of pandemic restrictions. We investigated any worsening of symptoms and/or the onset of new symptoms in many cognitive domains, namely, memory, attention, visuo-spatial functioning, orientation (temporal, spatial, and topographic), and language and any worsening in, and/or the onset of, BPSD, such as apathy, anxiety, depression, insomnia, delusion, hallucination, wandering, agitation, eating disorders, sundowning, and disinhibition compared to before lockdown. In this section, caregivers were asked to rate the severity of each symptom on a scale from 1 (very mild) to 4 (severe). Regarding patients’ autonomy, we investigated their independence in performing basic (ambulating, feeding, dressing, personal hygiene, continence, and toileting) and instrumental (ability to use the telephone, shopping, food preparation, housekeeping, laundry, mode of transportation, responsibility for their own medications, and ability to handle finances) activities of daily living (ADLs). Again, caregivers were asked to rate patients’ ADLs on a scale from 0 (dependence) to 1 (independence). The last part of the questionnaire (questions 128–157) focused on the caregivers’ experience of providing care over the lockdown period. We gathered information regarding any changes in their emotional experience and mood, self-efficacy in problem-solving, caregiving burden, and quality of life. The participants were asked to specify their level of agreement or disagreement on a symmetric agree–disagree scale for a series of statements (Likert scale) regarding lockdown compared to before and to rate some specific items from 0 (no changes) to 1 (very mild changes), 2 (mild changes), 3 (moderate changes), or 4 (severe changes).

### 2.3. Statistical Analysis

In this study, we collected many quantitative and qualitative data. We performed a deep descriptive analysis of these heterogeneous data sets by calculating measures of central tendency, variability, and frequency distributions. We performed three multivariable logistic regressions, including, as dependent variables, (1) cognitive worsening, (2) functional worsening, and (3) depression worsening (described by binary variables) and including, as independent variables, the patient’s education, gender, and age, as well as isolation and cohabitation. Then, we performed two multivariable logistic regressions, including, as dependent variables, (1) worsening of caregivers’ QOL and (2) worsening of caregiver’s burden and, as independent variables, the caregiver’s education, gender, age, and living situation (isolation and cohabitation).

## 3. Results

### 3.1. Demographic Data of the Study Sample and General Health Condition of the Patients

Two hundred and fifty-five caregivers of patients diagnosed with AD were randomly selected among those recorded in our Memory Clinic’s electronic archives. About 81.6% of them (*n* = 208) satisfied the inclusion/exclusion criteria of the protocol and gave their informed consent to participate in the study at the preliminary call. Eleven caregivers skipped the subsequent planned call, being excluded from the study, while 197 completed the survey. Table 1 shows the demographic data of the participants (patients and caregivers), including their education level, degree of kingship, cohabitation before the pandemic, and marital status during the lockdown, and the mean raw MMSE score for the patients (Table 1).

No cases of confirmed COVID-19 infection were reported among patients or caregivers. No patient had undergone quarantine or shown symptoms attributable to COVID-19. Only 4 (2.0%) patients and 8 (4.0%) caregivers underwent a nasopharyngeal swab, with a negative result. One caregiver reported some COVID-19-like symptoms, though the infection was excluded from medical/laboratory exams. Almost 40% (*n* = 78) of the caregivers reported changes in the patient’s general health condition and had to contact, at least once, their general practitioner (GP) and/or other specialized physicians (not neurologists) (Table 2).

The temporary closure of the Memory Clinic and/or the postponement of neurological visits led to significant difficulties in patient management in 19.8% (*n* = 39) of the cases.

### 3.2. Daily Organization of Caregiving During the Pandemic and Lockdown

Regarding patients’ living conditions before the pandemic, 110 (55.8%) of them did not receive any external assistance for their daily living, 44 (22.3%) needed half a day of care, and 43 (21.8) received caring 24 h a day/7 days a week. Over the pandemic period and lockdown, in 41 cases (20.8%), the relative caregiver moved in with the patient to provide care, becoming a cohabiting caregiver, whereas in 80 cases (40.6%), non-cohabiting caregivers increased the number of calls and video calls with the patient. Nevertheless, the frequency of the visits of non-cohabiting relatives decreased in 48.2% (*n* = 95) of the cases, and up to 12.2% (*n* = 24) of the patients did not meet any non-cohabiting relatives during the lockdown period. At the time of our survey, only 13.2% (*n* = 26) of patients lived alone at home. Regarding social activities, almost all the caregivers reported changes in daily habits, mostly in terms of practicing hobbies, leaving the house, and visiting family members. Ninety patients (45.7%) had to abandon hobbies and social interests that they regularly practiced before the pandemic, 101 (51.3%) reduced the frequency of leaving home, and up to 89 (45.1%) already no longer left home at all, before the lockdown restrictions were put in place. Only 10 (5.1%) caregivers reported a worsened economic condition that caused concern during the pandemic.

### 3.3. Effects of the Lockdown on the Patients

#### 3.3.1. Cognitive Functions

A large percentage of caregivers reported that the patients experienced a worsening in cognitive decline and a reduction in autonomy levels during the lockdown. Forty-eight patients (24%) developed new symptoms. Among them, 26 (54.2%) developed memory deficits, 22 (45.8%) developed language disturbances, 9 (18.7%) developed attention difficulties, 7 (14.6%) showed temporal and/or spatial disorientation, and 5 (10.4%) exhibited impaired executive functioning. Concerning pre-existing symptoms, 146 (74.1%) caregivers reported a worsening in at least one cognitive domain: 113 (57.4%) described a worsening in memory, 81 (41.1%) described a decline in attention, and 78 (39.6%) showed a worsening in linguistic skills. Temporal orientation also worsened in 70 (35.5%) patients, while spatial orientation worsened in 50 (25.4%), and navigational/topographical orientation worsened in 25 (12.7%). Most of the caregivers who described such worsening reported it as very mild or mild in terms of language (58 out of 78; 74.4%), attention (58 out of 81; 71.6%), and topographical orientation (19 out of 25; 76.0%) and mild or moderate in terms of memory (76 out of 113 caregivers; 67.3%), temporal orientation (46 out of 70 caregivers; 65.7%), and spatial orientation (33 out of 50 caregivers; 66.0%) (Figure 2).

#### 3.3.2. Levels of Functional Independence

Forty-one percent of caregivers (*n* = 82) stated that the lockdown severely impaired patients’ autonomy in general. In these cases, patients showed more difficulties than before, in housekeeping, preparing meals, and practicing leisure activities. Conversely, 115 (59.4%) reported no change or even an improvement in some skills, particularly using the telephone and practicing hobbies (Table 3).

#### 3.3.3. Awareness of COVID-19 Pandemic

Regarding patients’ awareness of the pandemic, 81.7% (*n* = 161) of them spontaneously spoke about it at least once a week, though only 52.8% (*n* = 104) spontaneously remembered that the lockdown was ongoing. Moreover, 18.8% (*n* = 37) of them did not remember which activities they could or could not do, and up to 47.2% (*n* = 93) of caregivers reported that the patient did not remember the pandemic at all. Lastly, only 5.1% (*n* = 10) of patients spontaneously adopted adequate behaviors (Table 4).

#### 3.3.4. Behavioral and Psychological Symptoms of Dementia

Regarding patients’ BPSDs, 183 (92.9%) caregivers reported the onset and/or the worsening of some neuropsychological symptoms, including apathy, anxiety, mood disorders, agitation, wandering, sundowning syndrome, delusions, hallucinations, sleep disturbances, eating disorders, or sexual problems. In 41.6% of cases (*n* = 82), these changes forced the caregiver to have a medical consultation with a neurologist, who suggested a pharmacological intervention (new therapies) and/or modification (dose changes) in about 32% (*n* = 63) of the cases. Figure 3 shows the numbers and percentages of caregivers reporting the onset of new BPSDs and modifications to pre-existing ones during the lockdown.

When analyzing each patient’s symptoms individually, it emerged that apathy, depression, anxiety, and agitation were the BPSDs that most frequently became worse during the lockdown. A wandering episode increase was reported by 24 caregivers, while in 12 patients, such symptoms appeared for the first time. Despite some caregivers highlighting an increase in sleep disorders, these symptoms remained substantially stable for most of them compared to before. Patients suffering from disinhibition and sexual behavior disorders remained almost stable, as well. Apathy, depression, sleep disorders (including insomnia), and wandering were the newly onset BPSDs more frequently reported by caregivers.

### 3.4. Effects of the Lockdown on Caregivers

#### 3.4.1. Mood and Emotions

Regarding caregivers’ mood changes and emotional experience, in general, they reported negative effects of the lockdown. Despite none of the interviewed caregivers having a diagnosis of depression or undergoing antidepressant therapy, 102 (51.8%) experienced at least one new depressive symptom or a worsening of some pre-existing symptoms during the lockdown (Table 5).

Our findings also highlighted that more than half of the caregivers (*n* = 100) experienced unpleasant emotions. In total, 61 % (*n* = 120) of the participants suffered from a lack of companionship more than before the pandemic, 48.2% (*n* = 95) felt a greater lack of emotional closeness with others, and 28.0% (*n* = 55) felt more alone in problem-solving. Furthermore, 39.1% (*n* = 77) of caregivers had less time for themselves, and 46.2% (*n* = 91) had fewer opportunities to engage in pleasant or useful activities for their well-being (Table 6).

Concerning the effect of the lockdown on the relationship between patients and caregivers, about 185 (93.9%) interviewed subjects reported the onset of new feelings towards the patient. In many cases (123 subjects, 62.4%), such feelings were positive, though they were negative for 49 (24.9%). Table 6 shows how frequently the caregivers felt these contrasting feelings in relation to the changes in caregiving and daily life habits during the lockdown (Table 7). Lastly, the quality of the relationship between patients and caregivers, in general, seemed to improve for most caregivers, 130 (66.0%), while it worsened for only 30 (15.2%).

When we asked the caregivers about their greatest concern related to the patients during the lockdown, most of the study sample (*n* = 126, 64.0%) answered that it was the development and/or worsening of BPSDs. Among the BPSDs studied throughout our survey, apathy was the most concerning. Ninety (45.7%) caregivers described apathy as a worrisome symptom, and 38 of them reported it as moderately to severely concerning. Depression, anxiety, and agitation were concerning, as well, for 77, 74, and 61 caregivers, respectively. While disturbances in the sexual sphere (including disinhibition) and eating disorders were considered less concerning, more than one-third of the caregivers facing hallucinations considered them severely worrisome (Figure 4).

#### 3.4.2. Caregiver Burden, Self-Efficacy in Problem-Solving, and Quality of Life

Over the lockdown period, 97 caregivers (49.3%) perceived caregiving as more burdensome than before, whereas 87 (44.2%) and 13 (7%) perceived a stable or a lower burden, respectively. Among those reporting a higher stress load in managing patients’ needs and facing the restrictions of the lockdown, 50 caregivers considered such an increase moderate to severe (Table 8).

Moreover, 79 (40.1%) subjects reported a higher frequency of new problems arising during the daily care tasks. Nevertheless, 43 (21.8%) considered lockdown a challenge and, at the same time, an opportunity to better solve existing problems related to disease management. Lastly, regarding caregivers’ self-efficacy, 31 (15.7%) felt less capable of problem-solving during the lockdown than before, whereas 28 (14.2%) felt more capable. The rest of the sample (138 subjects, 70.0%) did not notice any difference during the pandemic.

Compared to before, lockdown negatively affected caregivers’ quality of life (QOL) in 147 (74,6%) cases, and up to 57 (28.9%) and 14 (7.1%) reported this effect as moderate and severe, respectively (Table 9).

#### 3.4.3. Multivariable Logistic Regressions

The worsening of the patient’s cognitive, functional, and depression conditions were not found to be significantly associated with any predictors in the regression models. The worsening of the caregiver’s quality of life was found to be predicted by higher education levels (B = 0.96, SE = 0.48, beta = 1.1, *p* = 0.047).

## 4. Discussion

In this study, we investigated in depth, through an extensive telephone interview, caregivers’ mood, emotions, burden, and quality of life changes during lockdown and their perspectives on the effect of this complex phase of the pandemic outbreak in our country on AD patients’ cognitive state, behavior, and autonomy levels. Several previous studies investigated family caregivers’ experiences during the COVID-19 pandemic or explored caregiver burden and health conditions in dementia patients during lockdown through a telephone interview [10,11,12], but only one involved a large study sample like ours [13].

Our results extensively describe how the lockdown imposed profound changes in the daily organization of caregiving in AD and significantly contributed to a general worsening of patients’ cognitive and neuropsychiatric symptoms. These effects negatively affected the well-being and burden level of caregivers. Very often, the caregiver experienced an increased sense of loneliness and helplessness, developing anxiety and guilt regarding the lower or inadequate care provided [14], with negative effects on perceived QOL. Since neither patients nor caregivers in our sample showed symptoms of SARS-CoV-2, we may exclude any direct influence of the disease on the patients and caregivers.

Regarding the effects of the lockdown on patients’ cognitive status, most of the caregivers interviewed perceived a worsening, confirming previous data [11,12,13]. The changes in cognitive abilities mainly consisted of the exacerbation of pre-existing deficits in memory, attention, and language and new difficulties in learning and speech. Some questions in the survey investigated specific aspects of anterograde memory by asking about the patient’s awareness of the pandemic and their ability to learn new habits. We found that real awareness of the pandemic was rare in the study sample, since most of the patients showed severe difficulties in remembering why they could not leave their homes or receive visits. Interestingly, a severe decline in temporal–spatial and topographic orientation emerged as well. We hypothesize that mental and physical inactivity due to the limitations caused by lockdown might lead to a faster and more significant impairment in some cognitive functions. Socialization and social activities have repeatedly been reported to play a significant role in contrasting cognitive decline over time [15,16]. In line with this evidence, we confirm that the impossibility of attending day-care centers, participating in cognitive stimulation sessions, or engaging in outdoor or social activities promoting cognitive skills might contribute to cognitive impairment in AD. Moreover, the choice to limit the survey to the investigation of what happened over a very brief period (the lockdown lasted in Italy for about eight weeks) allowed us to speculate that such significant worsening might occur early, probably depending on the intensity of the cognitive deprivation due to lockdown.

Regarding the neuropsychological effects of the lockdown on the patients, we found that almost all caregivers reported some changes in patients’ behavior and mood [13]. We observed both new cases of reported affective–behavioral symptoms and a worsening in pre-existing behavioral disorders. The novel symptoms most frequently reported were apathy, depression, wandering, sleep, and eating disorders, while the worsening of pre-existing symptoms most frequently reported included apathy, deflection of mood, anxiety, and agitation. The lack of social contact may be considered one of the main factors associated with BPSDs worsening [17]. In addition to the diminished proximity of their family caregiver, during lockdown, almost half of the patients had to stop the social activities previously carried out. This evidence is in line with a recent review of the literature finding that the social isolation used to limit the spread of COVID-19 was associated with a range of documented neuropsychiatric consequences [18]. Our data suggest that decreased social contact and reduced daily activities outside the home could produce or exacerbate neuropsychiatric symptoms. Many patients went out less than before or stopped going out entirely at the start of the lockdown, thus reducing their possibility of maintaining even minimal social relations or motor activity. Furthermore, the high percentage of patients who lived alone and showed worsening neuropsychiatric symptoms also emphasizes the importance of a caregiver’s closeness. These data confirm previous evidence reporting the presence or exacerbation of neuropsychiatric symptoms as a sort of “deprivation syndrome” due to the lack of environmental input [19]. Patients need to interact with others, share their daily experiences, and receive emotional support and reassurance. Again, our results demonstrate how even a short period of intense social isolation such as that experienced due to the lockdown could lead to a widespread worsening in BPSDs in patients suffering from AD.

The general worsening of cognitive and behavioral status in the patients corresponded to reduced functional autonomy in daily life activities. Indeed, many caregivers reported reduced autonomy, especially in patients’ ability to use the telephone and household appliances and in performing domestic activities. Although disease progression could partially explain this worsening, the lockdown likely hastened the decline, contributing to an accelerated lack of autonomy in patients. However, it is noteworthy that in a few cases, the patient seemed to improve their abilities in some skills, showing more independence. For example, a small percentage of caregivers reported that patients improved in practicing hobbies. While a deterioration in many skills was expected, it was surprising to see improvement in some other skills. More time spent at home may have led to more time to practice certain activities with the caregiver’s help, with consequent performance enhancement. To our knowledge, our study is the first to have demonstrated these data, emphasizing once again the importance of skill stimulation in AD patients.

Regarding the emotional experience of caregivers during the lockdown, we also observed changes in line with the evidence of the literature suggesting that the role of the caregiver often deeply affects their emotional and psychological well-being [20,21]. About half of the caregivers in the study reported experiencing unpleasant emotions during the lockdown, greater loneliness, and a lack of companionship. They reported feeling alone in dealing with problems, lacking emotional closeness with others, and lacking time that they could spend alone. Together with the perceived burden increase, the worsening of caregivers’ emotional status led to a perceived impairment of physical and mental health. Indeed, in our survey, it emerged that a period of prolonged stress and increased emotional and caregiving load may compromise caregivers’ psychological well-being, resulting in an increased level of depression and stress, as previously reported [22]. Unsurprisingly, a third of caregivers reported the onset of new negative feelings toward their patients. These feelings likely contributed to a proportional increase in caregiver burden [23,24]. In our sample, about half of the caregivers reported experiencing a heavier care burden during the pandemic than before, with an exacerbation during the lockdown. Many caregivers felt less able than before to solve the problems encountered in their caregiver role and felt this role had become more difficult during the pandemic, with an increase in concerns due to patients’ clinical worsening. Our survey highlighted that some neuropsychiatric symptoms, namely, hallucinations, delusions, and agitation, resulted in the most concern for carers. Despite hallucinations and delusions having a low incidence in our sample, their nature and particular expression severely worried the caregivers; moreover, irritability, agitation, and aggressiveness were more frequently reported. It is easy to understand how these symptoms could negatively affect the emotional experience of caregivers and to imagine how stressful and frustrating it could be for a caregiver facing the same difficulties every day in the dramatic circumstances of a pandemic. That is why almost all the caregivers who completed the survey admitted a significant deterioration in their QOL during the lockdown. Regarding the perceived QOL, the multivariable logistic analysis that we performed highlighted a slightly, but interestingly, significant correlation: the higher the educational level of the caregiver, the worse their perception of their QOL. Ours is the first study finding such a correlation. It is possible that caregivers with a high education level might be more aware of the pandemic and therefore feel more concern about its effect on their and their relatives’ health. At the same time, they could also be more concerned about the effects of lockdown on their working life and economic conditions.

Despite the large number of questions, the ACQUA questionnaire was feasible since the vast majority of the subjects who satisfied the inclusion criteria to participate in the study completed the interview. Only 11 caregivers dropped out after giving their consent, skipping the main call planned. This high participation and adherence to the protocol was probably due to the great trust that the caregivers placed in the doctors and psychologists of the Memory Clinic and to their need to share what they experienced during the lockdown. This study has some potential limitations. First, since our goal was to investigate the caregivers’ perspectives, all the data concerning the patients’ cognitive and neuropsychiatric symptom onset and/or worsening cannot be considered an objective evaluation of the disease progression due to the lockdown. The emotional impact of the pandemic on caregivers may have partly affected their perception of the severity of patient symptoms. Therefore, verifying the concordance of these findings through face-to-face neuropsychological and neuropsychiatric evaluations of the patients could be interesting. Some issues with research samples and selection should also be considered. Furthermore, we cannot completely distinguish how much the lockdown itself, not directly related to the caregiving, influenced the caregivers, independently of its effects on the person living with dementia and on the caregiving. This is another limitation of the current study. The inclusion/exclusion criteria foresaw the caregivers’ willingness and availability to be contacted by phone twice. Hence, only those with sufficient time (about 60 min per call) participated in the study. Employing a telephone interview as a clinical data collection method might also be considered a potential limitation of our study. The telephone interview is an accepted and well-studied approach for quantitative data collection, although used less often than face-to-face interviews in qualitative research [25]. Nevertheless, many studies widely confirm the validity and advantages of such surveys in public health research [26]. Furthermore, the subjects enrolled in this research were chosen from the caregivers of AD patients; thus, the results obtained allow a description of what happened during the lockdown solely to such patients and not to others suffering from different neurodegenerative disorders. Furthermore, the heterogeneity of the qualitative data collected limited our statistical methods; thus, the reported results may only suggest how the lockdown due to COVID-19 influenced the participants. Moreover, the small size of our study sample might be responsible for the absence of some significant results in the multivariate logistic analysis performed. Another potential limitation of this study regards the exclusion from the survey of specific questions on caregivers’ and patients’ income status. In fact, the questionnaire included only one item that investigated general concern about the economic negative effects of the pandemic and lockdown on the patient and the caregiver. Future research should include this information, because it might greatly influence people’s everyday feelings and mood. Lastly, the absence of a control group might limit the validity of some of our results, but it was not possible to include a control group at the time, as the pandemic impacted everyone. This is why, again, future research should utilize the same semi-structured interview, with questions about changes over the same period, in a similar population after lockdown. Nevertheless, the employment of strict inclusion–exclusion criteria, the single-blind design, and the enrolment of a large sample of reliable caregivers ensured accurate data collection and should be considered the strengths of this study.

## 5. Conclusions

In this study, we investigated, through a telephone interview, the perspective of a large sample of Italian caregivers of AD patients on the effects of lockdown on the patient’s clinical status; the caregivers’ mood, emotions, burden, and QOL; and the caregiving during the COVID-19 pandemic outbreak. Our results, in line with previous research [11,12,13], have shown that in most cases, the extraordinary isolation and restrictions imposed by the lockdown severely changed the organization of caregiving, to the detriment of the psycho-physical well-being of the caregiver and the patient. Most of the family caregivers, who could no longer rely on help from others to assist the patient, faced the need to change their way of living during this period. Among those who did not live with the patient before the pandemic, some caregivers had to move in with the patient because of the sudden lack of any external (not familial) carer. In some cases, forced cohabitation with the patient brought out explicit negative thoughts toward the patient. In other cases, lockdown restrictions and limitations led the caregiver to further reduce their number of visits and limit their contact with the patient, with more frequent telephone calls bringing out feelings of guilt and sadness. In both cases, the impact of the new caregiving conditions on the caregivers led to the development of depressive symptoms and a worsening of their QOL. During the lockdown, there was a significant worsening in patients’ cognitive functioning and BPSDs, resulting in a further impairment in their autonomy levels. It is impossible to exclude the possibility that such changes could be partially due to the natural progression of AD over time, but the relatively brief period of observation (less than 3 months) suggests a main effect of the lockdown itself on patients. Surprisingly, some patients facing the new living conditions improved in their ability to carry out some tasks, showing some improved skills in everyday living.

The evident negative effects of the lockdown on patients heavily influenced caregivers, who often reported an increase in stress. These results are in line with those summarized in many recently published reviews, confirming caregivers’ fear and apprehension around COVID-19 infections, the sense of uncertainty and danger due to the restrictions imposed, the increased caregiving burden, loneliness, and isolation [27,28,29]. Our study has shown for the first time that many of these effects, already described in the literature, can occur with dramatic consequences following even a relatively short period of intense isolation and restriction. All this evidence confirms how important it is for a frail patient suffering from AD to be able to maintain an active social life, practice cognitively stimulating activities, and have a healthy relationship with their caregiver. Similarly, the effects of lockdown on caregivers demonstrate how these subjects should be considered “frail” as well. In the emergency of the pandemic, the high risk of contagion, and all the public health conditions that led many states around the world to face periods of lockdown, there were not immediate considerations of the short-term effects that the imposed restrictions would have in these patients. Our results might be useful in prompting institutions to pursue public health choices considering the psycho-physical health of AD patients and their caregivers when facing both ordinary and extraordinary conditions, such as a pandemic. At the same time, we hope that all the data that we collected in this survey might contribute to a better understanding of these patients’ need to be assisted, even remotely, in order to facilitate their participation in non-pharmacological activities of cognitive stimulation. The digital evolution observed in recent years has highlighted the value of cognitive stimulation and support for patients with dementia using information technology tools [30]. Likewise, our results confirm how much the caregiver’s integrity was at risk in exceptional caregiving conditions. Again, our data might suggest the importance of public health interventions dedicated to frail caregivers, by offering the possibility of remote psychological and/or medical assistance in more complex situations such as a pandemic and the opportunity to participate in assistance paths that can help caregivers to take back their lives after this type of exceptional situation.

## Figures and Tables

**Figure 1 ijerph-21-01622-f001:**
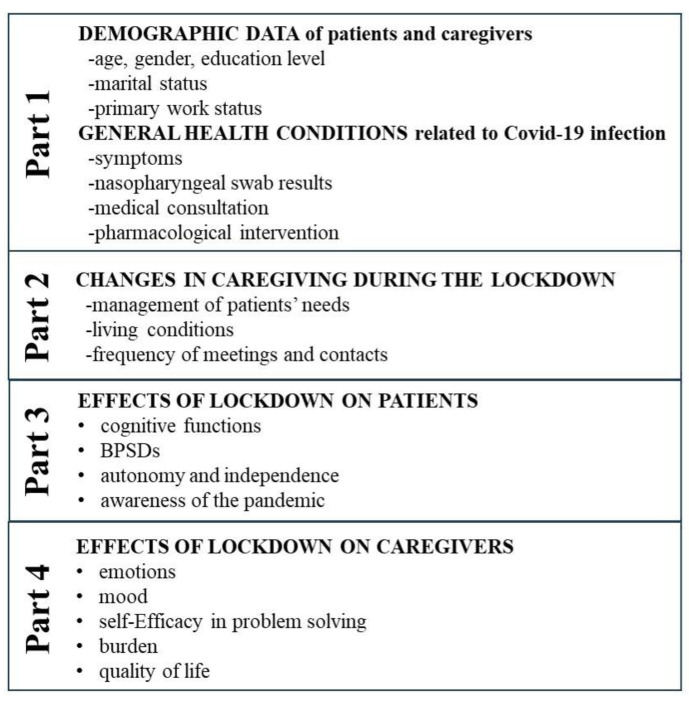
Alzheimer–COVID QUArantine (ACQUA) questionnaire. This figure shows the four areas of interest (PART 1–4) investigated throughout the questionnaire.

**Figure 2 ijerph-21-01622-f002:**
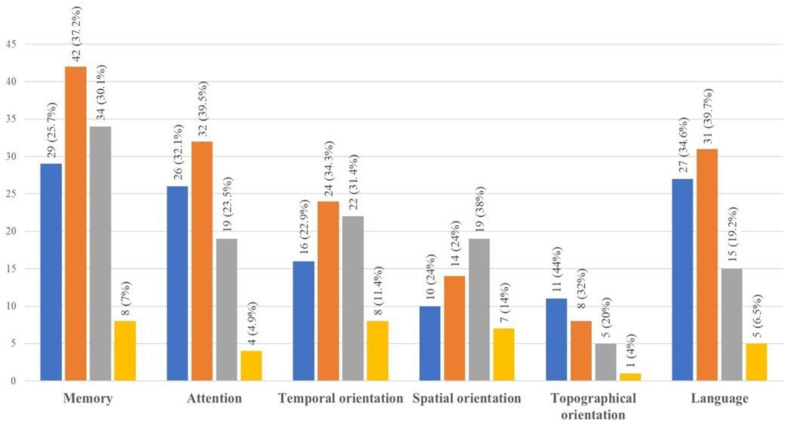
Worsening of pre-existing cognitive symptoms in the patients. The bars in the figure express the number (and percentages) of caregivers reporting a very mild (blue bar), mild (orange bar), moderate (gray bar), or severe (yellow bar) worsening in patients’ memory, attention, temporal, spatial and topographic orientation, and language. *Y*-axis, number of patients.

**Figure 3 ijerph-21-01622-f003:**
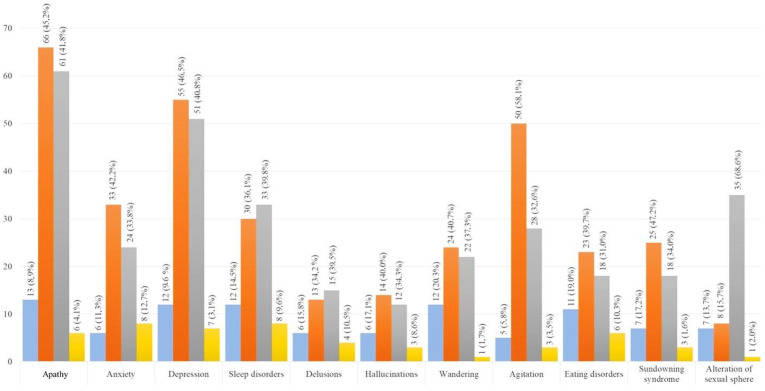
Changes in patients’ BPSDs during lockdown. The blue bar represents the onset of new neuropsychiatric symptoms, whereas the orange one, the gray one, and the yellow one represent, respectively, the worsening, stability, or improvement of pre-existing BPSDs. *Y*-axis, number of patients.

**Figure 4 ijerph-21-01622-f004:**
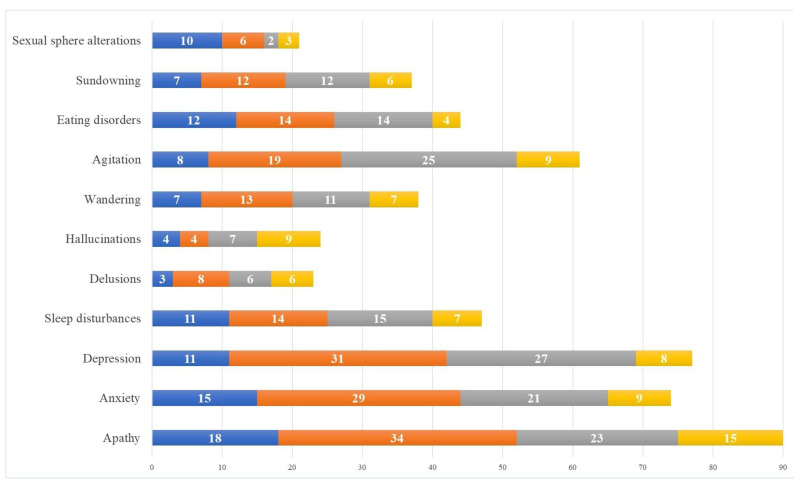
Caregivers’ concern about patients’ BPSDs: The figure shows the number of caregivers who reported very mild (blue bar), mild (orange bar), moderate (gray bar), or severe (yellow bar) concerns about the neuropsychiatric symptoms of the patients.

**Table 1 ijerph-21-01622-t001:** Demographic data and kinship degree of the participants. Education level in Italy: primary school (at least 5 years of school), lower secondary school/high school (at least 13 years of school), undergraduate/postgraduate (at least 17 years of school). SD, standard deviation.

**Patients**	**Total (*n* = 197)**
Age (years, mean ± SD) Sex (female, %)	77.2 ± 6.8 124 (63%)
Education (years, mean ± SD) -primary/lower secondary school -high school -undergraduate/postgraduate	9.7 ± 4.9 116 (58.9%) 69 (35%) 12 (6.1%)
MMSE (mean ± SD) Marital status (married, %) Widowed/separated (mean ± SD)	20.1 ± 2.4 131 (66.5%) 66 (33.5%)
**Caregivers**	Total (*n* = 197)
Age (years, mean ± SD) Sex (female, %)	60.1 ± 14.0 127 (64.5%)
Education (years, mean ± SD) -primary/secondary school -high school -undergraduate/postgraduate	13.9 ± 4.2 74 (37.6%) 93 (47.2%) 30 (15.2%)
Marital status (married, %)	151 (76.7%)
Widowed/separated (%) Cohabitant before lockdown (%)	46 (23.3%) 76 (37.1%)
**Degree of kinship**	*n*
Spouse	66 (33.5%)
Son/daughter	114 (57.8%)
Other	17 (8.7%)

**Table 2 ijerph-21-01622-t002:** Medical consultations during lockdown. This table shows the number and percentage of caregivers who needed to call a general practitioner and/or other physicians (not neurologists) for a consultation and treatment modification.

Need for Medical Consultation	*n*	%
General practitioner	61	31.0%
Other specialized physicians	7	3.6%
Both	10	5.1%
No need	129	65.5%

**Table 3 ijerph-21-01622-t003:** Changes in patients’ autonomy levels. This table shows the number and percentage of caregivers reporting an improvement, a worsening, or stability in patients’ autonomy in activities of daily living.

Activities of Daily Living	Improved	Stable	Worsened
Use of the telephone	15	144	38
Practicing hobbies	12	159	26
Housekeeping	8	152	37
Use of household appliances	4	180	13
Therapy management	2	170	25

**Table 4 ijerph-21-01622-t004:** Patients’ awareness of the pandemic. This table shows the number (and percentage) of patients remembering the pandemic, speaking about it, engaging in appropriate behaviors related to it, and remembering what they could/could not do.

Awareness of the Pandemic	Yes	Not
Remembers that the COVID-19 emergency is underway	104 (52.8%)	93 (47.2%)
Engages in appropriate behaviors without awareness	10 (5.1%)	187 (94.9%)
Remembers what they can/cannot do	160 (81.2%)	37 (18.8%)
Speaks spontaneously about confinement/restrictions	161 (81.7%)	36 (18.3%)

**Table 5 ijerph-21-01622-t005:** Depressive symptoms. This table shows the number and percentage of caregivers who reported the onset or the worsening of at least one depressive symptom during the lockdown.

Depressive Symptoms	*n*	%
None	95	48.2%
Low mood or sadness	68	34.5%
Feeling tearful	57	28.9%
Having less motivation or interests	48	24.4%
Feeling irritable and intolerant of others	28	14.2%
Feeling hopeless and helpless	25	12.7%
Feeling guilt-ridden	22	11.2%
Having low self-esteem	10	5.1%

**Table 6 ijerph-21-01622-t006:** Caregiver’s emotional experiences. This table shows the number and percentage of caregivers who reported any changes in the feelings and behaviors that they experienced during the lockdown compared to before.

Emotional Experience			
	Less than before	As before	More than before
Dealing with unpleasant emotions	5 (2.5%)	92 (46.7%)	100 (50.8%)
Lack of emotional closeness to people	4 (2.0%)	98 (49.7%)	95 (48.2%)
Lack of companionship	7 (3.5%)	70 (35.5%)	120 (61.0%)
Feeling alone in solving problems	12 (6.0%)	130 (66.0%)	55 (28.0%)
Time to be on their own	77 (39.1%)	102 (51.8%)	18 (9.1%)
Activities for their well-being	91 (46.2%)	60 (30.5%)	46 (23.3%)

**Table 7 ijerph-21-01622-t007:** Caregivers’ feelings towards the patients. This table shows the number and percentage of caregivers who reported any positive or negative feelings towards the patients and the frequency of such feelings during the lockdown.

Emerging Feelings Towards the Patients	Positive	Negative
No	*n* = 72 (36.5%)	*n* = 139 (70.5%)
Yes	*n* = 125 (63.5%)	*n* = 58 (30.5)
-Sometimes	*n* = 39 (19.8%)	*n* = 31 (15.8%)
-Often	*n* = 37 (18.8%)	*n* = 9 (4.5%)
-Very often	*n* = 40 (20.3%)	*n* = 14 (7.2%)
-Always/everyday	*n* = 9 (4.6%)	*n* = 4 (2.0%)

**Table 8 ijerph-21-01622-t008:** Changes in caregivers’ burden during lockdown. This table shows the number and percentage of caregivers who reported any change in caregiving burden compared to before.

Change in Caregivers’ Burden	*n*	%
No	87	44.2%
Less than before	13	6.6%
More than before	97	49.2%
-Very mildly	18	9.0%
-Mildly	29	14.6%
-Moderately	30	15.1%
-Severely	20	10.5%

**Table 9 ijerph-21-01622-t009:** Changes in caregivers’ QOL during lockdown. This table shows the number and percentage of caregivers who reported any change in their QOL compared to before.

Change in Caregivers’ Quality of Life	*n*	%
No	50	25.4%
Yes	147	75.6%
-Very mild	38	19.3%
-Mild	38	19.3%
-Moderate	57	28.9%
-Severe	14	7.1%

## Data Availability

All the materials and data collected for this study are stored in the Department of Human Neuroscience at the “Sapienza” University of Rome. For more information, please contact the corresponding author by email.

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
