# Peer review of "Neuropsychological Effects of the Lockdown Due to the COVID-19 Pandemic on Patients with Alzheimer’s Disease and Their Caregivers: The “ACQUA” (Alzheimer–COVID QUArantine Questionnaire) Study"

_ijerph, 2024, doi:10.3390/ijerph21121622_

Round 1

Reviewer 1 Report (Previous Reviewer 2)

Comments and Suggestions for Authors

First of all, I appreciate your work and the effort to improve it. We proceed to make the considerations

INTRODUCTION

Lines 80-83 “On the best of our knowledge, the present study is the first attempt to deeply investigate, throughout an extensive telephone interviews, the impact of lockdown and other restrictive measures on patients with AD and their family caregivers.”

Lines 339-342 “As far as we know, this is the first study that deeply investigated using a telephone 339 survey, the caregivers’ mood,…”

These comments are unnecessary; they are a value judgement.

METHODS

2.1. Participants

Lines 95-96“Second, s/he must be available by tele-94 phone at designated times and, in the investigators’ opinion, must have adequate literacy to complete the protocol-specified questionnaire.”

It would be inappropriate to speak of opinion in a scientific study, especially when the level of education is to be a quantitative variable in the study.

2.2. Alzheimer Covid QUArantine (ACQUA) questionnaire

The questionnaire will be also attached as an annex.

RESULTS

It would be necessary to present the results in tables.

Although some socio-demographic variables (gender, age, marital status, employment) are known, no results are reflected in relation to these parameters.

The variables and the results of the logistic regression model should be accompanied by a table. What variables have been assessed on the caregiver: gender, age, marital status, apart from educational level, or any special characteristics of the caregiver? Perhaps the severity of the patient's changes?

RESULTS

The variables and the results of the logistic regression model should be accompanied by a table. What variables has been assessed on the caregiver - gender, age, marital status, or any special caregiving characteristics, apart from educational level? Perhaps the severity of the patient's symptomatology, or cognitive status?

Line 228.

You should take out the word opinion and put in another expression, similar to ‘perceived by them’.

DISCUSSION

Line 347-349

You should avoid value judgements such as:

“These effects negatively affected the 347 well-being and burden level of the caregivers who felt the consequences of anti-social pandemic measures and restrictions, leading to a deterioration of QOL.”

The antisocial aspect of quarantine is a matter of controversy and includes aspects and objectives not addressed in this paper.

The discussion is too long

Author Response

Comments 1:

First of all, I appreciate your work and the effort to improve it. We proceed to make the considerations

INTRODUCTION

Lines 80-83 “On the best of our knowledge, the present study is the first attempt to deeply investigate, throughout an extensive telephone interviews, the impact of lockdown and other restrictive measures on patients with AD and their family caregivers.”

Lines 339-342 “As far as we know, this is the first study that deeply investigated using a telephone 339 survey, the caregivers’ mood,…”

These comments are unnecessary; they are a value judgement.

Response 1:

Thank you very much for your observations. We edited the text accordingly.

Comments 2:

METHODS

2.1. Participants

Lines 95-96“Second, s/he must be available by tele-94 phone at designated times and, in the investigators’ opinion, must have adequate literacy to complete the protocol-specified questionnaire.”

It would be inappropriate to speak of opinion in a scientific study, especially when the level of education is to be a quantitative variable in the study.

Response 2:

Thank you very much for your observations. We agree with the reviewer and we edited the text accordingly.

Comments 3:

2.2. Alzheimer Covid QUArantine (ACQUA) questionnaire

The questionnaire will be also attached as an annex.

Response 3:

Thank you very much for your suggestion. We added the questionnaire as requested in the "supplementary materials".

Comments 4:

RESULTS

It would be necessary to present the results in tables.

Although some socio-demographic variables (gender, age, marital status, employment) are known, no results are reflected in relation to these parameters.

Response 4:

Thank you very much for your observations. We now added a table with all these data.

Comments 5:

The variables and the results of the logistic regression model should be accompanied by a table. What variables have been assessed on the caregiver: gender, age, marital status, apart from educational level, or any special characteristics of the caregiver? Perhaps the severity of the patient's changes?

Response 5:

Thank you very much for your suggestion. Since our statistical analysis showed no significative results, decided to only describe the results in the text not employing another table (the manuscript already contains 13 figures/tables). However, If you consider it mandatory, we might add the table requested.

Comments 6:

Line 228.

You should take out the word opinion and put in another expression, similar to ‘perceived by them’.

Response 6:

Thank you very much for your observations. We agree with the reviewer, and we edited the text accordingly.

Comments 7:

DISCUSSION

Line 347-349

You should avoid value judgements such as:

“These effects negatively affected the 347 well-being and burden level of the caregivers who felt the consequences of anti-social pandemic measures and restrictions, leading to a deterioration of QOL.”

The antisocial aspect of quarantine is a matter of controversy and includes aspects and objectives not addressed in this paper.

The discussion is too long

Response 7:

Thank you very much for your observations. Again, we agree with the reviewer, and we edited the text accordingly and revised the Discussion (see DISCUSSION and CONCLUSION).

Reviewer 2 Report (New Reviewer)

Comments and Suggestions for Authors

In the current manuscript “Neuropsychological effects of the lockdown due to the Covid-19 pandemic on patients with Alzheimer’s disease and their caregivers: The “ACQUA” (Alzheimer-Covid QUArantaine questionnaire)” the authors want to provide insights into the neuropsychological consequences of the covid-19 lockdown on people living with Alzheimer’s disease and the change in care burden, emotions, and quality of life for their caregivers. Caregivers were investigated using a semi-structured telephone questionnaire covering demographic information, and retrospective assessment of general health conditions and changes in caregiving and effects on patients and on caregivers due to the lockdown. The descriptive summary of the data indicated that both the people living with dementia and caregivers felt that many psychological symptoms worsened.

The general research idea to investigate the impact of the lockdown on cognitive and emotional changes and everyday life functioning in individuals living with dementia and their caregivers is highly relevant and compelling. However, the way this was operationalized and measured in the current manuscript precludes drawing any conclusions on this issue. As a result, I have major concerns suggesting this manuscript for publication. In the following I am only highlighting the three major concerns.

My first major concern is that the aim of the study does not align with the hypotheses, study design or data analyses. The study aims to investigate the neuropsychological effects of the lockdown on individuals living with Alzheimer’s disease and their caregivers. However, the first hypothesis is that the pandemic effects neuropsychological change in patients leading to increased care needs, which is moderated by reliance on care. The second hypothesis is that the neuropsychological changes lead to changes in the caregivers. Thus, for the first hypothesis the outcome measurement seems to be care needs, which is not specified in the aim of the study. Additionally, the second hypothesis does not align with the aim of the study either, since it assumes that the lockdown only has an effect on caregivers through symptom changes in the patient. This completely disregards the effect of the lockdown itself on psychological changes in caregivers, despite studies reporting this effect (e.g., Carbone et al., 2021). However, neither the research questions nor hypotheses can be answered with the current study design, as there is not control group. Of course, it was not possible to include this control group at the time as the pandemic impacted everyone. Therefore, it is uncertain whether the reported changes are due to the effects of lockdown as proposed, or due the natural progression of Alzheimer's disease and possible retrospective report bias.  One possibility would have been to conduct the same semi-structured interview, with questions about changes over the same time period in the same (similar) population after the lockdown to disentangle changes, which are common when a person is asked twice and in retrospect manner from those, which are linked to the conditions of the pandemic. Furthermore, the data does not test the research question or the hypotheses. The analyses merely consist of a descriptive summary of the data and three regression analyses. The regression analyses only included stable characteristics (education, gender and age), which are not affected by the lockdown. Therefore, these regressions do not provide any information regarding the effect of the lockdown. If the authors wanted to investigate this effect, they should have used variables which change during the lockdown as independent variables,  such as social isolation, mobility (Cinar et al., 2022; Ingram, Hand, & Maciejewski, 2021). Furthermore, these relationships should be controlled for cohabitation and symptom severity (Aragón et al., 2022).

My second major concern is that it remains completely unclear how the variables are analysed and scored. First, it is stated that both quantitative and qualitative are collected. However, the authors do not mention how these are disentangled and analysed separately. It almost seems as if only the quantitative data is analysed. For example, it is unspecified whether the worsening or onset of new symptoms was based on a predetermined list of symptoms or open-ended questions. If there was a predetermined list, there should also be an answering option for remaining stable, to avoid biased results of change. If it was an open-ended question, information should be provided on how this was qualitatively analysed (see e.g., Kallio et al., 2016 for qualitative data analysis guidelines). Furthermore, when the authors do specify how the variables are coded, the scoring does not always seem appropriate. For instance, for the rating of change the scoring seems to be: 0 for no change, 1 for improvement and 2 for worsening during the lockdown. However, this should be worsening (0), no change (1) and improvement (2) to analyze this in a linear fashion. Additionally, the semi-structured interview was conducted for this study, but the authors do not provide any justification for the inclusion or exclusion of certain items. The authors should also investigate and report the reliability of the measurement. In general, as the questionnaire and the different scales are so vaguely described, it is difficult to imagine what would have been possible with the available data. Furthermore, qualitative data analyses is more than a description of answering patterns. Unfortunately, this manuscript currently reads like a summary of these descriptive patterns – hence, claiming to combine qualitative and quantitative analyses in this manuscript is not convincing.

Finally, with the current data design, I find it difficult to disentangle the impact of the lockdown on (a) the person living with dementia and (b) the caregivers independently from each other, as the data are only collected from the subjective and retrospective perspective of the caregivers. Their self-perception of the changes and effects of the pandemic on the person living with dementia is not independent on how much they were affected by the lockdown themselves. As no objective measures are taken into consideration, this is a major limitation of the current study and limits, in addition to the points mentioned above, the validity of the results.

References

1)         Aragón, I., Flores, I., Dorman, G., Rojas, G., Sierra Sanjurjo, N., & O’Neill, S. (2022). Quality of life, mood, and cognitive performance in older adults with cognitive impairment during the first wave of COVID 19 in Argentina. International Journal of Geriatric Psychiatry, 37(1). https://doi.org/ https://doi.org/10.1002/gps.5650

2) Cinar, N., Sahin, S., Karsidag, S., Karali, F. S., Ates, M. F., Gonul, O., Okluoglu, T., Eren, F., Bulbul, N. G., & Okuyan, D. Y. (2022). Neuropsychiatric effects of Covid-19 pandemic on alzheimer’s disease: A comparative study of total and partial lockdown. The Medical Bulletin of Sisli Etfal Hospital, 56(4), 453. https://doi.org/DOI:10.14744/SEMB.2022.40326

3) Ingram, J., Hand, C. J., & Maciejewski, G. (2021). Social isolation during COVID19 lockdown impairs cognitive function. Applied Cognitive Psychology, 35(4), 935-947. https://doi.org/https://doi.org/10.1002/acp.3821

4)  Kallio, H., Pietilä, A. M., Johnson, M., & Kangasniemi, M. (2016). Systematic methodological review: developing a framework for a qualitative semistructured interview guide. Journal of advanced nursing, 72(12), 2954-2965. https://doi.org/https://doi.org/10.1111/jan.13031

Comments on the Quality of English Language

In general fine - sometimes a bit wordy

Author Response

Comments 1:

In the current manuscript “Neuropsychological effects of the lockdown due to the Covid-19 pandemic on patients with Alzheimer’s disease and their caregivers: The “ACQUA” (Alzheimer-Covid QUArantaine questionnaire)” the authors want to provide insights into the neuropsychological consequences of the covid-19 lockdown on people living with Alzheimer’s disease and the change in care burden, emotions, and quality of life for their caregivers. Caregivers were investigated using a semi-structured telephone questionnaire covering demographic information, and retrospective assessment of general health conditions and changes in caregiving and effects on patients and on caregivers due to the lockdown. The descriptive summary of the data indicated that both the people living with dementia and caregivers felt that many psychological symptoms worsened.

The general research idea to investigate the impact of the lockdown on cognitive and emotional changes and everyday life functioning in individuals living with dementia and their caregivers is highly relevant and compelling. However, the way this was operationalized and measured in the current manuscript precludes drawing any conclusions on this issue. As a result, I have major concerns suggesting this manuscript for publication. In the following I am only highlighting the three major concerns.

My first major concern is that the aim of the study does not align with the hypotheses, study design or data analyses. The study aims to investigate the neuropsychological effects of the lockdown on individuals living with Alzheimer’s disease and their caregivers. However, , which is moderated by reliance on care. The second hypothesis is that the neuropsychological changes lead to changes in the caregivers. Thus, for the first hypothesis the outcome measurement seems to be care needs, which is not specified in the aim of the study. Additionally, the second hypothesis does not align with the aim of the study either, since it assumes that the lockdown only has an effect on caregivers through symptom changes in the patient. This completely disregards the effect of the lockdown itself on psychological changes in caregivers, despite studies reporting this effect (e.g., Carbone et al., 2021). However, neither the research questions nor hypotheses can be answered with the current study design, as there is not control group. Of course, it was not possible to include this control group at the time as the pandemic impacted everyone. Therefore, it is uncertain whether the reported changes are due to the effects of lockdown as proposed, or due the natural progression of Alzheimer's disease and possible retrospective report bias.  One possibility would have been to conduct the same semi-structured interview, with questions about changes over the same time period in the same (similar) population after the lockdown to disentangle changes, which are common when a person is asked twice and in retrospect manner from those, which are linked to the conditions of the pandemic. Furthermore, the data does not test the research question or the hypotheses. The analyses merely consist of a descriptive summary of the data and three regression analyses. The regression analyses only included stable characteristics (education, gender and age), which are not affected by the lockdown. Therefore, these regressions do not provide any information regarding the effect of the lockdown. If the authors wanted to investigate this effect, they should have used variables which change during the lockdown as independent variables, such as social isolation, mobility (Cinar et al., 2022; Ingram, Hand, & Maciejewski, 2021). Furthermore, these relationships should be controlled for cohabitation and symptom severity (Aragón et al., 2022).

Response 1:

Thank you very much for your observations. As we stated in the introduction we hypothesized: 1) the pandemic and its restrictions might worse patients’ cognition and behaviour and 2) the pandemic and its restrictions might worse caregivers’ mental health and increase their burden. In line with these two hypotheses, our study aimed to investigate the caregivers’ perspective on a) the effects of lockdown on cognition, behaviour, and autonomy levels of their AD patients and b) its impact on the emotions and moods of the caregivers themselves. To do that we proposed a specific telephone survey (the ACQUA questionnaire) that deeply investigates the subjective caregivers’ perspective. Caregivers’ perspective was the object of our research. Our study design excluded a control group because it was not possible to include it during the pandemic (we agree with the reviewer). We hope future research might include a control group as you suggested. Accordingly, we added that as a limit of the study (please see DISCUSSION).

With regard to the effect of the lockdown on the caregivers, our second hypothesis did not assume that the lockdown had an effect on caregivers exclusively through symptom changes in the patient. As stated in the introduction, we hypothesized that lockdown restrictions might negatively influence family caregivers’ physical and mental health, mainly increasing the caregiving burden. In fact, the last part of the ACQUA questionnaire investigated emotion, mood, self-efficacy in problem-solving, burden and quality of life of the caregivers in general (related to/not related to the caregiving) during the lockdown.

With regard to the statistical analysis, the data obtained were extremely heterogeneous (qualitative/quantitative). That is why we performed only a descriptive analysis at first. Then we proposed limited multivariable logistic regressions, but in agreement with the reviewer we decided to perform a deeper analysis as you suggested including more variables. We revised the statistics accordingly including living situation, isolation and cohabitation as well as symptoms worsening (please see MATERIALS AND METODS). Unfortunately, these further analyses did not find out significative results.

With regard to the effect of the AD progression over time, again we agree with the reviewer. It is impossible to exclude that cognitive and behavioural worsening depended on the natural history of the disease, but the relatively brief period of observation (less than 3 months) should exclude it. We edited the CONCLUSION accordingly.

Comments 2:

My second major concern is that it remains completely unclear how the variables are analysed and scored. First, it is stated that both quantitative and qualitative are collected. However, the authors do not mention how these are disentangled and analysed separately. It almost seems as if only the quantitative data is analysed. For example, it is unspecified whether the worsening or onset of new symptoms was based on a predetermined list of symptoms or open-ended questions. If there was a predetermined list, there should also be an answering option for remaining stable, to avoid biased results of change. If it was an open-ended question, information should be provided on how this was qualitatively analysed (see e.g., Kallio et al., 2016 for qualitative data analysis guidelines). Furthermore, when the authors do specify how the variables are coded, the scoring does not always seem appropriate. For instance, for the rating of change the scoring seems to be: 0 for no change, 1 for improvement and 2 for worsening during the lockdown. However, this should be worsening (0), no change (1) and improvement (2) to analyze this in a linear fashion. Additionally, the semi-structured interview was conducted for this study, but the authors do not provide any justification for the inclusion or exclusion of certain items. The authors should also investigate and report the reliability of the measurement. In general, as the questionnaire and the different scales are so vaguely described, it is difficult to imagine what would have been possible with the available data. Furthermore, qualitative data analyses is more than a description of answering patterns. Unfortunately, this manuscript currently reads like a summary of these descriptive patterns – hence, claiming to combine qualitative and quantitative analyses in this manuscript is not convincing.

Response 2:

Thank you very much for your comments. We agree with the reviewer. The ACQUA questionnaire includes more than 150 open and closed questions, with dichotomous or Likert-scale answers. We obtained many heterogeneous quantitative and qualitative data. This is an important limit of the study as highlighted in the discussion, but we updated our statistics as suggested above. Furthermore, we now added the QUESTIONNAIRE as an annex (supplementary material).

Comments 3:

Finally, with the current data design, I find it difficult to disentangle the impact of the lockdown on (a) the person living with dementia and (b) the caregivers independently from each other, as the data are only collected from the subjective and retrospective perspective of the caregivers. Their self-perception of the changes and effects of the pandemic on the person living with dementia is not independent on how much they were affected by the lockdown themselves. As no objective measures are taken into consideration, this is a major limitation of the current study and limits, in addition to the points mentioned above, the validity of the results.

Response 3:

Thank you again for your comment. We agree with the reviewer. We confirm the caregivers’ perspective is the “core” of our study. Our aim was to investigate it over a very short period by considering their subjective and retrospective perspective. We considered the caregiver as a witness of what happened during the lockdown. We did not want to quantify and measure cognitive decline in the patients since we did not employ any telephone-administered neuropsychological scales. Anyway, we updated the DISCUSSION accordingly.

Reviewer 3 Report (New Reviewer)

Comments and Suggestions for Authors

The study investigated the effects of COVID-19 pandemic lockdowns on Alzheimer's patients and their caregivers. The authors recorded their data through a questionnaire administered through a telephone interview. The questionnaire was conceived by the authors and comprised of 157 open and closed questions, with dichotomous or Likert-scale answers. It was found that Alzheimer's patients were negatively affected by the lockdown, and as a consequence caregiver burden and stress was heightened. The authors recommended that more assistance be provided to caregivers of such patients during future pandemics. The findings of the study supported that of previous similar reviews, suggesting that the ACQUA questionnaire conceived by the authors can be a functional tool for the assessment of patient and caregiver strain in future studies. 

Comments:

There are other studies that have performed telephone interviews with caregivers and Alzheimer's patients affected by the lockdown, e.g. 

https://doi.org/10.1177/07334648211036399

https://doi.org/10.4103/aian.aian_439_21

https://doi.org/10.3390/healthcare12100970

The authors should carefully account for these studies and emphasize the novelty of their own methodology/findings. 

Comments on the Quality of English Language

Comments:

Minor revisions necessary to improve grammar and readability of sentences throughout the article, e.g: "The main outcomes were 1) any changes in..." (Line 23) can be "The main outcomes were 1) changes in..." and "This study might contribute to understand the impact of lockdown..." (Line 28) can be "This study might contribute to understanding..." and so on and so forth. More extensive English proof-reading is recommended.

Author Response

Comments 1:

The study investigated the effects of COVID-19 pandemic lockdowns on Alzheimer's patients and their caregivers. The authors recorded their data through a questionnaire administered through a telephone interview. The questionnaire was conceived by the authors and comprised of 157 open and closed questions, with dichotomous or Likert-scale answers. It was found that Alzheimer's patients were negatively affected by the lockdown, and as a consequence caregiver burden and stress was heightened. The authors recommended that more assistance be provided to caregivers of such patients during future pandemics. The findings of the study supported that of previous similar reviews, suggesting that the ACQUA questionnaire conceived by the authors can be a functional tool for the assessment of patient and caregiver strain in future studies. 

Comments:

There are other studies that have performed telephone interviews with caregivers and Alzheimer's patients affected by the lockdown, e.g. 

https://doi.org/10.1177/07334648211036399

https://doi.org/10.4103/aian.aian_439_21

https://doi.org/10.3390/healthcare12100970

The authors should carefully account for these studies and emphasize the novelty of their own methodology/findings. 

Response 1:

Thank you very much for the suggestions. We revised the discussion and updated referencies as suggested.

Comments 2:

Comments on the Quality of English Language

Comments:

Minor revisions necessary to improve grammar and readability of sentences throughout the article, e.g: "The main outcomes were 1) any changes in..." (Line 23) can be "The main outcomes were 1) changes in..." and "This study might contribute to understand the impact of lockdown..." (Line 28) can be "This study might contribute to understanding..." and so on and so forth. More extensive English proof-reading is recommended.

Response 2:

Thank you very much again. We edited the manuscript accordingly.

Reviewer 4 Report (New Reviewer)

Comments and Suggestions for Authors

In this study, authors used telephonic conversations with caregivers of Alzheimer's disease patients to identify burden, quality of life, and stress levels during Covid19 pandemic and lockdown. Also, they examined the effect of the pandemic on the psychological state of the patients. 

1. Demographic data of the patients and caregivers can be presented as a detailed table (Table 1) for better visualization. This should also include education levels in detail- highschool, undergraduate, and postgraduate degree-as later the data is significant. 

2. Line 167-168: +/- SD needs to be removed.

3. Figure 2 & 3: Define the y-axis. Also, the colors of the bar should be included with the graph for reference.

4.  Did the emerging feeling towards the patient correlate with the patient's symptoms of deterioration or merely due to the pandemic?

5. What's the relationship status (kinship) between the patient and caregiver and did this account for the burden levels?

6. Does income status affect the burden on caregivers?

Author Response

Comments 1:

In this study, authors used telephonic conversations with caregivers of Alzheimer's disease patients to identify burden, quality of life, and stress levels during Covid19 pandemic and lockdown. Also, they examined the effect of the pandemic on the psychological state of the patients. 

1. Demographic data of the patients and caregivers can be presented as a detailed table (Table 1) for better visualization. This should also include education levels in detail- highschool, undergraduate, and postgraduate degree-as later the data is significant. 

Response 1:

Thank you very much for the suggestion. The manuscript alreadi contains 12 figures/tables/schemes. That is why we decided at first, to not include more tables and describe the demographic data only in the text. Nevertheless, we agree with the reviewer and we decided to include all the demographic data (including education levels etc.) in a table (Table 1). The other tables has been renumbered accordingly.

Comments 2:

2. Line 167-168: +/- SD needs to be removed.

Response 2:

Done as requested.

Comments 3:

3. Figure 2 & 3: Define the y-axis. Also, the colors of the bar should be included with the graph for reference.

Response 3:

Thank you very much for the comment. Y-axis defines the number of patients in both the figures. The colors of the bars are included in the figures’ captures. We now edited them accordingly.

Comments 4:

4. Did the emerging feeling towards the patient correlate with the patient's symptoms of deterioration or merely due to the pandemic?

Response 4:

Thank you very much for your question. The majority of the caregivers reported positive feelings towards the patient and the quality of the relationship with them seemed to improve during the lockdown. The carers’ main concern regarded patients’ health during lockdown. That is why we suppose the emerging feelings towards the patient mainly depend on patient’s symptoms deterioration (please see the DISCUSSION).

Comments 5:

5. What's the relationship status (kinship) between the patient and caregiver and did this account for the burden levels?

Response 5:

Again, thank you very much for your question. We update the text and the Table 1 highlightening the degree of kinship. No significant correlation found between burden and relationship status were found (see RESULTS and DISCUSSION).

Comments 6:

6. Does income status affect the burden on caregivers?

Response 6:

Thank you very much for your question. Only one question of the survey asked about “general concern” on patient’s and caregiver’s economic situation due to the pandemic and lockdown, but we did not investigate more their income status. Future researches might investigate it deeply.  We added that as a potential limit of the study (please see the DISCUSSION).

This manuscript is a resubmission of an earlier submission. The following is a list of the peer review reports and author responses from that submission.

Round 1

Reviewer 1 Report

Comments and Suggestions for Authors

Review of manuscript no. ijerph-3069972

“Neuropsychological effects of the lockdown due to the Covid-19 pandemic on patients with Alzheimer’s disease and their caregivers: the “ACQUA” (Alzheimer-Covid QUArantine questionnaire) study”.

This study reports the results of a telephone survey of people caring for a family member suffering from mild to moderate Alzheimer’s dementia. The survey was conducted in August/September 2020, after the lockdown restrictions were suspended (that is, after May 2020). A semi-structured questionnaire was completed with questions specifically developed for this study. The study found that nearly all of the caregivers admitted a significant deterioration of their quality of life whereas their feelings toward their dependents were more varied with more feeling more positively toward the person (63.5%) than those reporting feeling more negatively toward their dependent (30.5%).  The area of greatest concern to the caregivers was the worsening of the behavioural and psychological symptoms associated with their dependents’ condition.

The data presented are essentially descriptive and offer no explanations for the different pattern of responses.  It might have been useful to examine the dimensions of change reported – using some form of multivariate analysis like principle components or factor analysis, to reduce the data to common sources of variance and to include some information about the characteristics of those reporting a relative worsening of problems compared with those not reporting such worsening (e.g. nature of family relationship, whether or not they shared the same household before the pandemic, whether the caregiver was the only caregiver and whether they had an occupation or not.  Equally there seems to be no account of whether other forms of communication beyond face-to-face interactions took place and how often such contacts arose. In general I found that the recounting of descriptive data without any further analysis limited the value of the study .   There is also little reference to the many studies already published in this field, including in this journal  (a start could be made with one of the earliest reviews of this research – e.g. (a) Suárez-González, A., Rajagopalan, J., Livingston, G., & Alladi, S. (2021). The effect of COVID-19 isolation measures on the cognition and mental health of people living with dementia: A rapid systematic review of one year of quantitative evidence. EClinicalMedicine39  Sept. 2021,  (b) Bailey, C., Guo, P., MacArtney, J., Finucane, A., Swan, S., Meade, R., & Wagstaff, E. (2022). The experiences of informal carers during the COVID-19 pandemic: a qualitative systematic review. International Journal of Environmental Research and Public Health, 19(20), 13455, and (c) Baldassar, L., Nguyen, T. N. M., Jones, B., Stevens, C., Krzyzowski, L., Lozeva, S., Marino, S., Du Plooy, M.G.C., Eldridge, J., Almeida, O.P. & Ghosh, M. (2023). The impacts of COVID-19 restrictions on care-givers of people with cognitive impairment and their support needs: a mixed-methods systematic review. Ageing & Society, 1-31).

Comments on the Quality of English Language

Finally the English language quality of the article could be improved by asking for expert overview in order to eliminate various awkward expressions phrases and words in the text.

Author Response

Please see the attached file;

Reviewer 2 Report

Comments and Suggestions for Authors

First of all, I appreciate your work. We proceed to make the considerations

ABSTRACT.

Methods: The method is not clearly stated What type of test was used? There is no statistical analysis

“The present study aimed to investigate cognitive, psychological and behavioural effects of the lockdown in a population of Alzheimer’s disease patients living at home through an extensive semi-structured telephone interview with their family caregivers. We also investigated the effects of lockdown on caregivers’ emotional experiences and stress load”.

Results: No reference is made to any numerical value. If there is no indication of which method has been used the severity of the patient's condition is a value judgement, not a result.

“Our data showed the lockdown severely impaired patients’ cognition and autonomy and increased behavioural and psychological symptoms of dementia”.

MANUSCRIPT

METHODS

2.1. Participants

Lines 92-94, 96-97

“First, an eligible caregiver must be a reliable study partner in frequent contact with the patient (at least 15 hours per week before the lockdown)”

“Lastly, they must have accompanied the patient for a visit 96 at least once over the twelve months before the beginning of the Covid-19 pandemic”

Could you explain why these criteria were used?

2.2. Alzheimer Covid QUArantine (ACQUA) questionnaire

As the questions in the questionnaire are variable, they should be detailed in a supplementary figure.  Not having a list of the variables that have been selected makes it difficult to interpret the results. Figure 1 is not adequately clarifying.

2.3. Statistical analysis (lines 127-129).

The work is purely descriptive and does not allow for some assertions made in the results. The heterogeneity of the sample and variables is not a justification. See this work, carried out on a similar topic, in a multicentre study, in the same country, in the same period, in which one of the authors participated. Not cited in the bibliography not cited in the bibliography.

Rainero I. et al. The Impact of COVID-19 Quarantine on Patients With Dementia and Family Caregivers: A Nation-Wide Survey and Family Caregivers: A Nation-Wide Survey. Front. Aging Neurosci. (2020). 12:625781. doi: 10.3389/fnagi.2020.625781.

RESULTS

As we said above not having a list of the variables that have been selected makes it difficult to interpret the results.

For example, marital status is chosen because it is intended to assess the variable social isolation, or because it influences cognitive reserve? Then, why is not asked about the level of education, which is of importance in the progression or not of cognitive decline? This question was included in the ACQUA of aforementioned work.

Changes in health conditions are referred to. It is relevant to know what these have been

Changes in medication, if relevant, why do we talk about? Talking about “changes in some therapies” is imprecise (line 155).

How the caregivers assessed the variation in cognitive functions such as a worsening in memory, in attention, in linguistic skill, special orientation, etc.? What questions did they have to answer to make such a specialised consideration of the patient's condition? If they were trained to do so, it should have been indicated in the inclusion criteria. Is it understood that the questionnaires were provided to caregivers by trained Neuropsychologists? So how did the caregivers rate them? The information that would be obtained would be quantitative, but it is referred to qualitatively.

“Figure 2. Worsening of pre-existing cognitive symptoms in the patients. The bars in the figure express the number (and percentages) of caregiver reporting a very mild (blue bar), mild (orange bar), moderate (grey bar) or severe (yellow bar) worsening in patients’ memory, attention, temporal, spatial and topographic orientation and language.

In the line 243 you say

“Regarding the caregivers’ mood changes and emotional experience, in general they reported significant negative effects of the lockdown.

line 357-358).

“Our data, in fact, confirmed that decreased social contact and reduced daily activities outside the home could produce or exacerbate neuropsychiatric symptoms” (DISCUSSION).

What statistical methodology has been used to support this assertion? This would be in line with the need for a statistical method to support the statement made in the abstract about the severity of the changes suffered by patients.

How do we know mathematically, for example, what factor has influenced the loss of quality of life of the caregiver? Did previous variables such as the caregiver's age, sex, or marital status influence their worsening quality of life? What relationship does the caregiver's condition have with the patient's worsening? In any case, perhaps it is not a problem of heterogeneity, but rather a small sample size, which makes it difficult to advance in the analysis

Consequently we think that It is essential for this work:

-clarify which variables have been selected,

 -apply a statistical treatment that allows us to talk about significance, severity and correlations.
